# Analysis of the saliva metabolic signature in patients with primary Sjögren's syndrome

Zhen Li[1], Yue Mu[1], Chunlan Guo[1], Xin You[2], Xiaoyan Liu[3], Qian Li[1]*, Wei Sun[3]*

1 Department of Stomatology, Peking Union Medical College Hospital, Chinese Academy of Medical Sciences, Beijing, China, 2 Department of Rheumatology and Clinical Immunology, Peking Union Medical College Hospital, Chinese Academy of Medical Sciences, Beijing, China, 3 Core Facility of Instrument, Chinese Academy of Medical Sciences, School of Basic Medicine, Institute of Basic Medical Sciences, Peking Union Medical College, Chinese Academy of Medical Sciences, Beijing, China

* liqianpumch@126.com (QL); sunwei@ibms.pumc.edu.cn (WS)

## Abstract

### Background

The saliva metabolome has been applied to explore disease biomarkers. In this study we characterized the metabolic profile of primary Sjögren's syndrome (pSS) patients and explored metabolomic biomarkers.

### Methods

This work presents a liquid chromatography-mass spectrometry-based metabolomic study of the saliva of 32 patients with pSS and 38 age- and sex-matched healthy adults. Potential pSS saliva metabolite biomarkers were explored using test group saliva samples (20 patients with pSS vs. 25 healthy adults) and were then verified by a cross-validation group (12 patients with pSS vs. 13 healthy adults).

### Results

Metabolic pathways, including tryptophan metabolism, tyrosine metabolism, carbon fixation, and aspartate and asparagine metabolism, were found to be significantly regulated and related to inflammatory injury, neurological cognitive impairment and the immune response. Phenylalanyl-alanine was discovered to have good predictive ability for pSS, with an area under the curve (AUC) of 0.87 in the testing group (validation group: AUC = 0.75).

### Conclusion

Our study shows that salivary metabolomics is a useful strategy for differential analysis and biomarker discovery in pSS.

**Data Availability Statement:** All relevant data are within the paper and its Supporting Information files.

**Funding:** The author(s) received no specific funding for this work.

**Competing interests:** The authors have declared that no competing interests exist.

# Background

Sjögren's syndrome (SS) is a multisystem autoimmune disease characterized by hypofunction of salivary and lacrimal glands and possible systemic multiorgan manifestations [1]. SS is a complex and heterogeneous disorder characterized by different clinical subsets [2], such as thrombocytopenia [3], leukopenia [4], lymphoma [5], interstitial pneumonia [6], renal tubular acidosis [7], and peripheral nerve involvement [8]. In the era of personalized medicine, new biomarkers are required to diagnose SS early, to define different disease subsets, to direct patients' clinical management and to monitor disease progression [9].

Saliva is a complex fluid containing a variety of metabolites, proteins, mRNAs, DNAs, enzymes, hormones, antibodies, antimicrobial constituents, growth factors and other molecules that may be associated with disease phenotypes [10, 11]. Compared to blood, saliva is easily, rapidly and noninvasively collected [12, 13]. Salivary gland dysfunction in primary Sjögren's syndrome (pSS) patients inevitably leads to changes in salivary components, and the pathogenesis of the disease can thus be inferred from changes in saliva. Many previous studies have focused on saliva proteomic, transcriptomic and genomic biomarkers for pSS [14, 15]; however, a sensitive and specific biomarker in this fluid source has not yet been available.

Metabolites, which are important indicators of physiological or pathological states, can provide information for the identification of early and differential markers of disease and help in understanding disease occurrence and progression [16]. Overall, saliva metabolomics has become a useful strategy for identifying biomarkers for oral squamous cell carcinoma [17, 18], periodontal diseases [19], oral lichen planus [20], oral leukoplakia [21] and cardiovascular disease [22]. In addition, a few metabolomics studies have recently reported salivary [23–25] sample biomarkers for pSS diagnosis. Both Mikkonen JJ and Herrala M used nuclear magnetic resonance (NMR) spectroscopy to analyze the saliva metabolome, and the levels of choline, taurine, alanine, glycine, phenylalanine and proline were found to be higher in patients with pSS than in controls [24, 25]. In another study, the saliva of 12 pSS patients was assessed by gas chromatography–mass spectrometry (GC–MS), and glycine, tyrosine, uric acid and fucose were found to be downregulated [23].

Ultra-performance liquid chromatography coupled with high-resolution mass spectrometry (UPLC-HRMS) is considered an appropriate technique for metabolomics studies, especially for large-scale untargeted metabolic profiling [26]. However, the use of the UPLC-HRMS method for detecting salivary metabolites in pSS patients has not been reported. The major objective of our study was to discover potential salivary biomarkers to distinguish patients with pSS from healthy controls by UPLC-HRMS.

# Materials and methods

## Ethics statement

The investigation protocol was approved by the Institutional Review Board of the Institute of Basic Medical Sciences, Chinese Academy of Medical Sciences, Project No:047–2019. Verbal consent forms indicating agreement to serve as saliva donors for the experiments were obtained from all study subjects before participation in the study (S1 File).

## Sample collection

This trial focused on pSS. A total of 32 pSS and 38 healthy control saliva samples were collected and randomly divided into two groups: the test group included 20 pSS patients and 25 healthy human adults, and the validation group included 12 pSS patients and 13 healthy controls. The participants were recruited from July 2019 to January 2020 from the Department of

Rheumatology and Clinical Immunology, Peking Union Medical College Hospital. These groups did not include subjects with any acute condition. All 32 patients with pSS were recruited according to the 2016 American College of Rheumatology (ACR)/European League Against Rheumatism (EULAR) classification criteria for pSS [1] by a rheumatologist to confirm the diagnosis according to medical history, clinical manifestations and examination and that pSS was not secondary to other connective tissue diseases (CTDs). The detailed demographics and disease pathology of the patients are shown in Table 1.

Unstimulated whole saliva samples were collected by having the participants spit for 15 min into a container between 9:00 and 11:00 a.m. The subjects were requested to refrain from eating, drinking, smoking or oral hygiene procedures for at least 1.5 hr prior to unstimulated whole saliva collection and to rinse their mouth with water prior to sample collection. The samples were transported to the laboratory on ice and centrifuged at $1500 \times g$ for 15 min at 4˚C to remove shed cell debris and residue, and then the supernatants were frozen immediately and stored at -80˚C until analysis [23].

## Sample preparation

For saliva metabolomics, acetonitrile (200 μL) was added to each saliva sample (200 μL), and the mixture was vortexed for 30 s and centrifuged at $14,000 \times g$ for 10 min. The supernatant was dried under vacuum and reconstituted with 200 μL of 2% acetonitrile. The quality control (QC) sample was a pooled saliva sample prepared by mixing aliquots of all samples across the two groups to be analyzed and was therefore globally representative of the whole sample set. The QC samples were injected every ten samples throughout the analytical run to provide a set of data from which method stability and repeatability could be assessed.

## UPLC-HRMS analysis

UPLC-HRMS analyses of samples were conducted using a Waters ACQUITY H-class LC system coupled with an LTQ-Orbitrap Velos pro mass spectrometer (Thermo Fisher Scientific, MA. USA). Saliva metabolites were separated with a 17-min gradient using a Waters HSS C18 column (3.0× 100 mm, 1.7 μm) at a flow rate of 0.3 mL/min. Mobile phase A was 0.1% formic acid in $H_2O$, and mobile phase B was acetonitrile. Saliva metabolites were separated with a gradient as follows: 0–2 min, 2% solvent B; 2–5 min, 2–55% solvent B; 5–15 min, 55–100% solvent B; 15–20 min, 100% solvent B; 20–20.1 min, 100–2% solvent B; and 20.1–29 min, 2% solvent B. The column temperature was set at 45˚C. Full MS acquisition was performed from 100 to 1000 m/z at a resolution of 60 K. The automatic gain control (AGC) target was $1 \times 10^6$, and the

**Table 1. Demographics and clinical characteristics of the pSS patients and healthy control subjects.**

| Characteristic | Test group | | Validation group | |
|---|---|---|---|---|
| | pSS patients | HCs | pSS patients | HCs |
| Number of samples | 20 | 25 | 12 | 13 |
| Age ($\bar{x} \pm s$) | 39.7±10.9 | 40.6±12.8 | 41.4±12.2 | 44.1±12.7 |
| Sex (F/M) | 17/3 | 23/2 | 10/2 | 12/1 |
| Anti-SSA/Ro- antibody positivity | 16/20 | | 9/11 | |
| Anti-SSB/La- antibody positivity | 3/20 | | 1/11 | |
| Immunoglobulin G positivity | 12/20 | | 6/12 | |
| Labial gland biopsy positivity | 14/17 | | 9/10 | |

pSS- primary Sjögren's syndrome, HCs- healthy controls, SSA- Sjögren's syndrome-related antigen A, SSB- Sjögren's syndrome-related antigen B, $p < 0.05$ for Wilcoxon test for pSS patients and HCs.

maximum injection time (IT) was 500 ms. UPLC targeted-MS/MS data were acquired at a resolution of 15 K with an AGC target of $5 \times 10^4$, a maximum IT of 250 ms, and an isolation window of 3 m/z. The collision energy was optimized as 20, 40, 60 or 80 for each target with higher-energy collisional dissociation (HCD) fragmentation. The injection order of the saliva samples was randomized to reduce experimental bias.

## Statistical analysis

Raw data files were processed by Progenesis QI (Version 2.0, Nonlinear Dynamics) software based on a previously published identification strategy, which included sample alignment, peak picking, peak grouping, deconvolution, normalization by total compounds, and final information export [27]. To make features more comparable, further data preprocessing, including missing value estimation, log2 transformation and Pareto scaling, was carried out using MetaAnalyst 3.0 (http://www.metaboanalyst.ca). Variables missed in 50% or more of all samples were removed from further statistical analysis. Nonparametric tests (Wilcoxon rank-sum test) were employed to evaluate the significance of variables. False discovery rate (FDR) correction was used to estimate the chance of false positives and correct for the testing of multiple hypotheses. The adjusted p value (FDR) cutoff was set as 0.05. Pattern recognition analysis (principal component analysis, PCA; orthogonal partial least squares discriminant analysis, OPLS-DA) was performed using SIMCA 14.0 (Umetrics, Sweden) software. The differential variables selected met the following three conditions: (i) adjusted $p < 0.05$; (ii) fold change between the two groups >2; and (iii) variable importance in OPLS-DA projection (VIP) above 1. Exploratory ROC analysis and external biomarker validation were carried out using the "Biomarker discovery" module with the MetaboAnalyst 3.0 platform.

Potential biomarkers were further evaluated by receiver operating characteristic (ROC) analysis. The area under the ROC curve (AUC), an efficient indicator of model performance, was used as a metric to assess the sensitivity and specificity of the biomarkers. AUC values > 0.9 indicate high reliability of the model, 0.7 to 0.9 indicate moderate reliability, 0.5 to 0.7 indicate poor reliability, and AUC $\leq$ 0.5 suggests that the model prediction is not better than chance.

## Metabolite annotation and pathway analysis

Metabolic pathways and predicted metabolites in the pathways were analyzed using the "Mummichog" algorithm based on the MetaboAnalyst 3.0 platform. Mummichog is a program written in python for analyzing data from high-throughput, untargeted metabolomics, bypassing the tedious and challenging metabolite identification. It leverages the organization of metabolic networks to predict functional pathways directly from feature tables and generate a list of tentative metabolites annotations through functional activity analysis. Metabolite annotation was further determined from the exact mass composition, from the goodness of isotopic fit for the predicted molecular formula and from MS/MS fragmentation comparing hits with databases (HMDB http://www.hmdb.ca/), thus qualifying for annotation at MSI level II using Progenesis QI. For endogenous metabolites lacking a chemical formula, an accurate molecular mass was determined based on the calculated isotopic features and ion adducts. Detailed methods are listed in the Supplementary Methods.

## Results

The workflow of our study is shown in Fig 1. Metabolite variation and pathway regulation associated with pSS were explored based on an analysis of metabolic profile differences between pSS and healthy controls. Potential biomarkers for pSS were further explored based on differential metabolites and validated using 10-fold cross-validation or external validation.

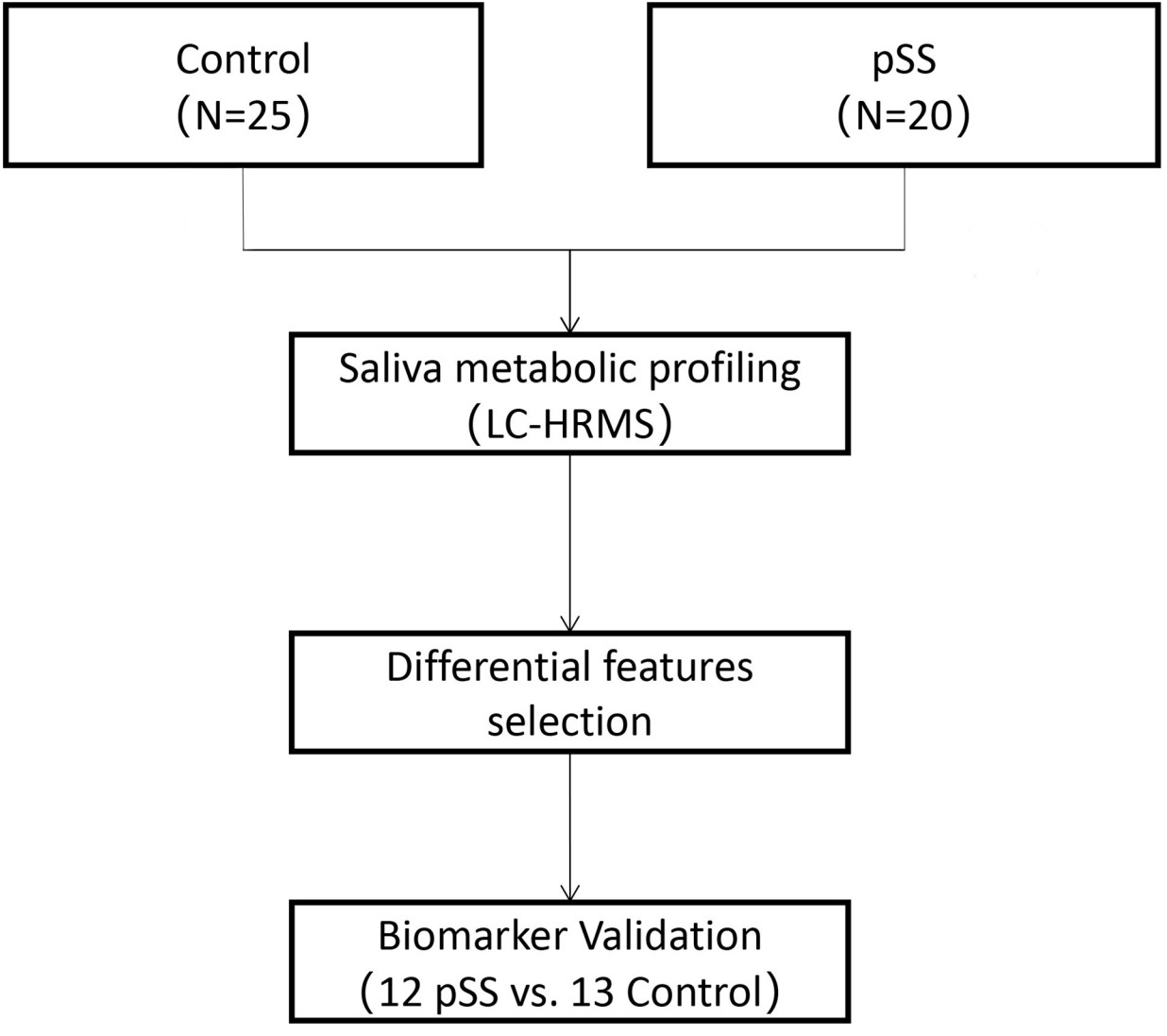

**Fig 1. The workflow of our study.**

### Characteristics of the participants

All the pSS patients included in the study had either ocular and/or oral symptoms, and all were positive for one or more autoantibodies. The autoimmune antibody profiles revealed anti-Ro-/Sjögren's syndrome A-antibody (anti-Ro/SSA) positivity in 73.5% (25/31) of the participants and anti-La-/Sjögren's syndrome B-antibody (anti-La/SSB) positivity in 11.8% (4/31) of the participants. In total, 27 pSS patients underwent labial gland biopsy, with positive results in 23. Detailed information is shown in Table 1.

### Quality control

To reduce experimental variations from the sampling process, standard sampling procedures, including sampling time and sampling processing, were performed by professionals. To ensure the repeatability and consistency of the metabolomics data, a quality control (QC) sample consisting of all samples was injected every ten or twelve sample injections. Principal component

analysis was performed to evaluate the variation in samples. A good cluster of QC samples indicated good stability of the analytical platform (S1A Fig).

## Distinction of pSS patients from healthy controls by saliva metabolomics

Based on UPLC-HRMS analysis, 2887 variables were quantified and analyzed from pSS and control saliva samples after QC filtering.

First, PCA was performed to explore the tendency of metabolic profiling variations between the healthy controls and pSS subjects (Fig 2A) and suggested apparent discrimination. Next, the OPLS-DA model achieved better separation for differential metabolite selection (R2X = 0.413, R2Y = 0.932, Q2 = 0.46, CV-ANOVA, $p$ = 0.00070, Fig 2B). Permutation tests (100 times) were performed to confirm the stability and robustness of the supervised models presented in this study (intercept of Q2 = - 0.261, S1B Fig).

Overall, 676 features with a P value < 0.05 (351 upregulated and 325 downregulated features) were submitted for pathway enrichment analysis using the Mummichog algorithm. The results showed significant enrichment ($p < 0.05$) of several pathways, including tryptophan metabolism, tyrosine metabolism, carbon fixation, and aspartate and asparagine metabolism, etc. Metabolites involved in tryptophan metabolism and aspartate and asparagine metabolism were upregulated. Metabolites involved in carbon fixation were downregulated (Fig 3).

Further annotation of the top discriminatory features ($p < 0.05$; $FC > 2$; $VIP > 1$) determined by MS/MS evaluation identified 38 significantly differentially abundant metabolites (S1 Table; relative intensity plotted as a heatmap in Fig 4).

Notably, the number of metabolites derived from amino acids was upregulated, and many of them were dipeptides. These metabolites include phenylalanyl-alanine, tryptophyl-isoleucine, tyrosyl-phenylalanine, asparaginyl-valine, aspartyl-isoleucine and tyrosyl-hydroxyproline. Additional differential metabolites related to purine metabolites, included 8-hydroxyadenine and oxypurinol. All the metabolites were present at higher levels in pSS.

To evaluate the diagnostic accuracy of these differential metabolites for pSS, a predictive model for patient classification was constructed using each identified metabolite. All 38 metabolites had a potentially useful diagnostic value, with an AUC above 0.7, and 30 metabolites had a good diagnostic value, with an AUC above 0.8. Among the saliva metabolites, aspartyl-isoleucine had the highest area under the ROC curve for the diagnosis of pSS (0.88, $p<0.001$), and

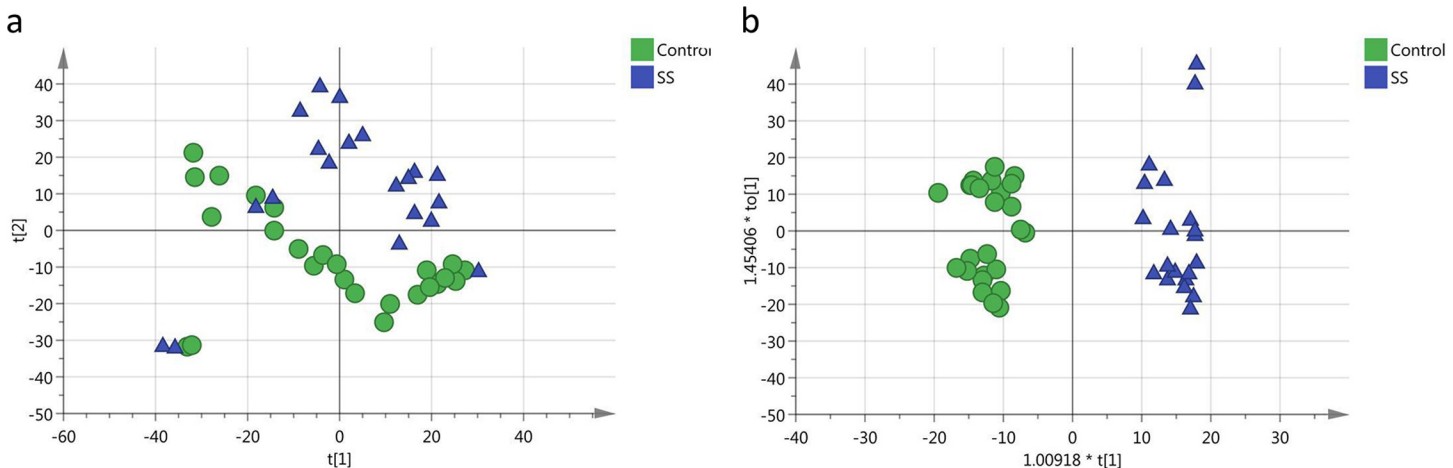

**Fig 2. Comparison between pSS subjects and healthy controls.** (a) Score plot of the PCA-X model between pSS subjects (blue) and healthy controls (green). (b) Score plot of the OPLS-DA model between pSS subjects (blue) and healthy controls (green).

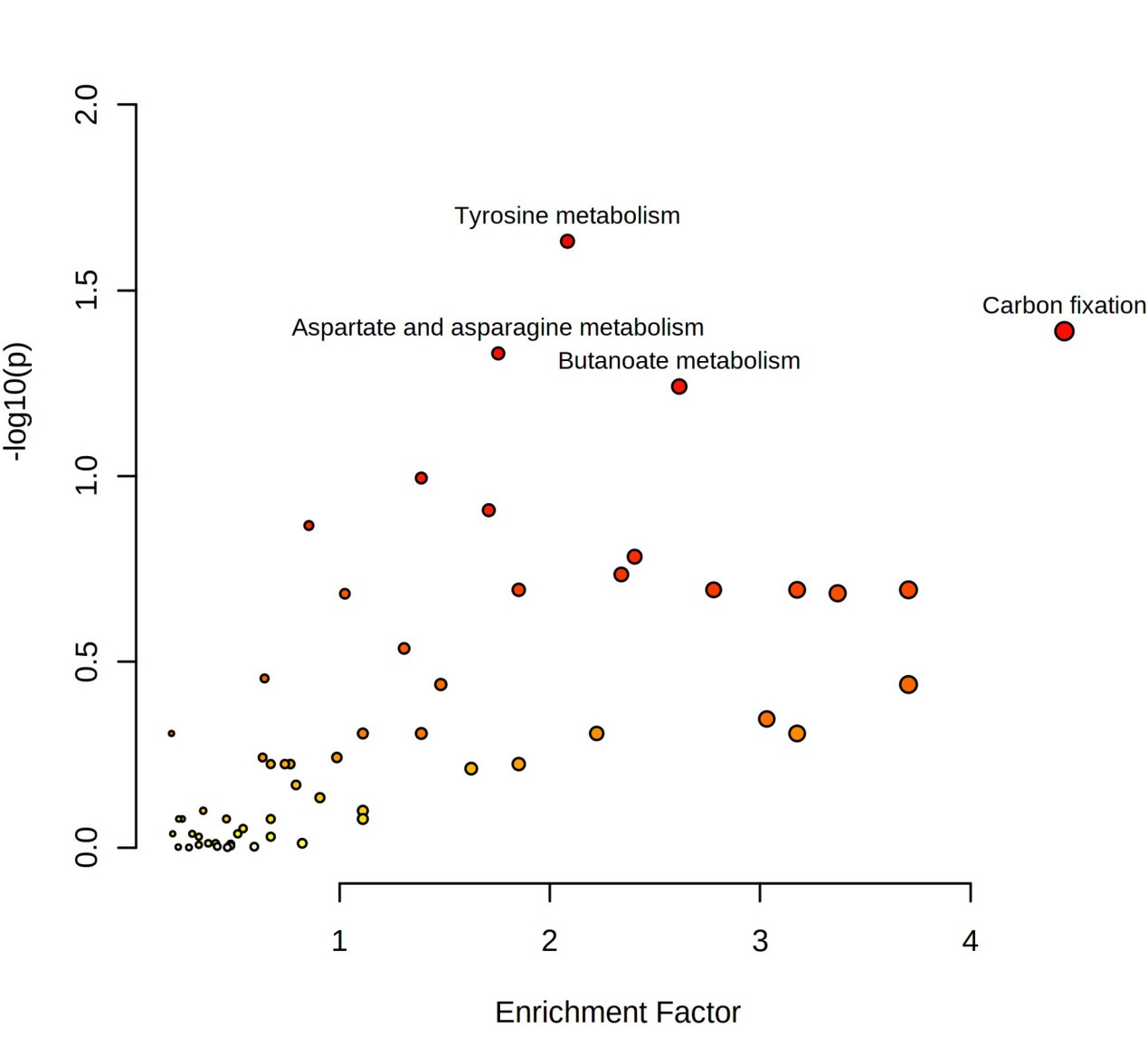

**Fig 3. Pathway enrichment analysis of differential metabolites of pSS and healthy controls.**

both the sensitivity and specificity reached 80%. Phenylalanyl-alanine for the classification of pSS patients and controls had an AUC of 0.87 for the test group, a sensitivity of 80%, and a specificity of 84%. To limit overfitting, the training set was subjected to 10-fold cross-validation to evaluate the stability and generalization of the metabolite. Further external validation using an independent sample set was carried out and achieved good performance with an AUC of 0.75, sensitivity of 75% and specificity of 85% (S2 Table and Fig 5).

## Discussion

Saliva is produced primarily by the major salivary glands, and the impaired secretion of the salivary glands can be described by the metabolites of saliva, which is an excellent diagnostic

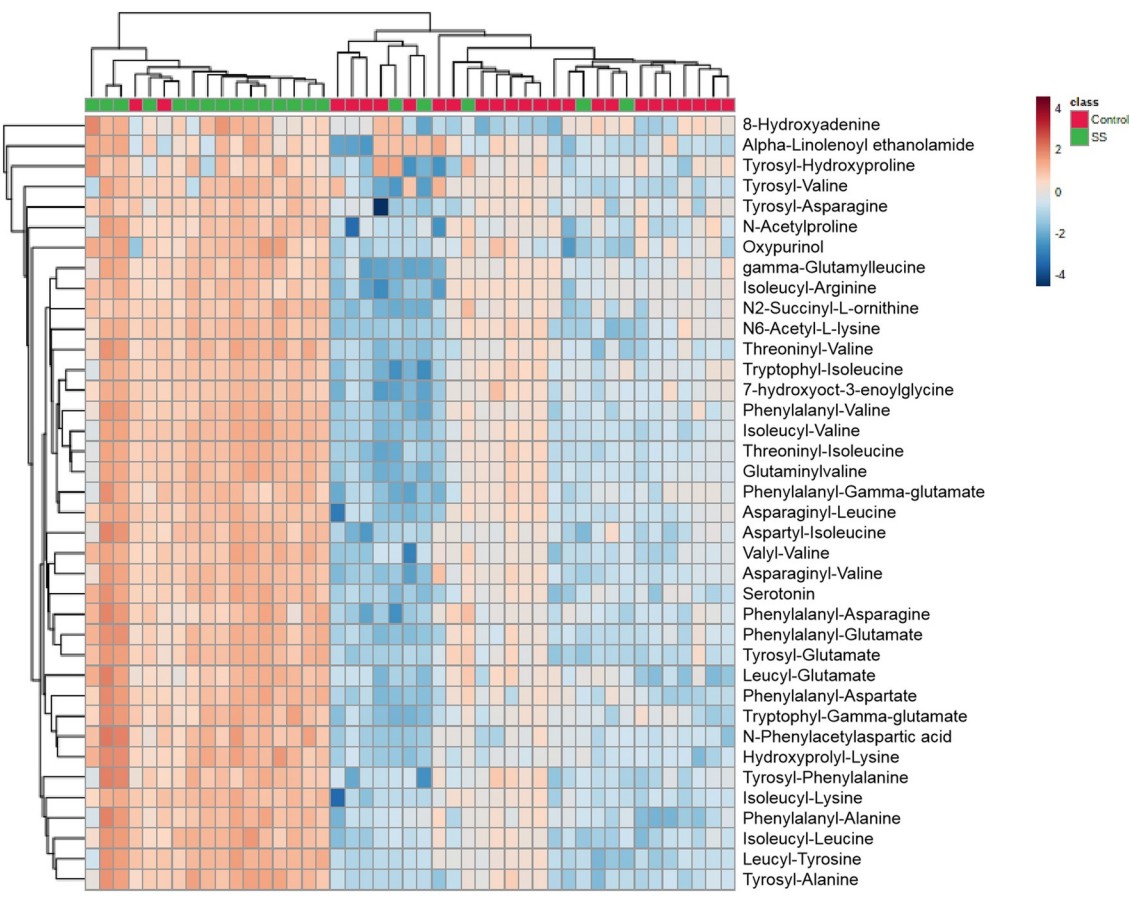

**Fig 4. Relative intensity of differential metabolites in pSS subjects and healthy controls.**

medium for clinical diagnostics [28]. In the present study, we conducted a comprehensive characterization of the saliva metabolome of 32 pSS patients and 38 healthy subjects. Our results suggest that the saliva metabolome can be used to differentiate pSS patients from healthy controls.

Whole saliva is a complex fluid containing a variety of substances, and changes in the levels of these substances are related to the pathogenesis of various diseases. However, the function of the salivary glands in patients with pSS is severely weakened, resulting in the volume of saliva being too small. Patients' whole saliva up to 2 ml was collected for analysis over time to obtain a pre-normalization of saliva.

Reliable identification of features distinguishing biological groups in salivary metabolite fingerprints requires the control of total metabolite abundance. Normalizations to one index values (for example, creatinine for urine metabolomics), osmolality, and total compound (useful MS signals) were the commonly used normalization techniques for overcoming sample variability in fluids metabolomics. Previous researches have compared the effect of different normalization method on urine metabolomics study. It showed that the best performance was normalization to the total compound. The approach does not only correct for different urinary output, but it also accounts for injection variability (REF: Evaluation of dilution and normalization strategies to correct for urinary output in HPLC-HRTOFMS metabolomics.). In present study, we normalized the data using the method of "Normalize to useful MS signals" to reduce sample variation. However, it cannot completely solve the problem of pre-normalization. It is

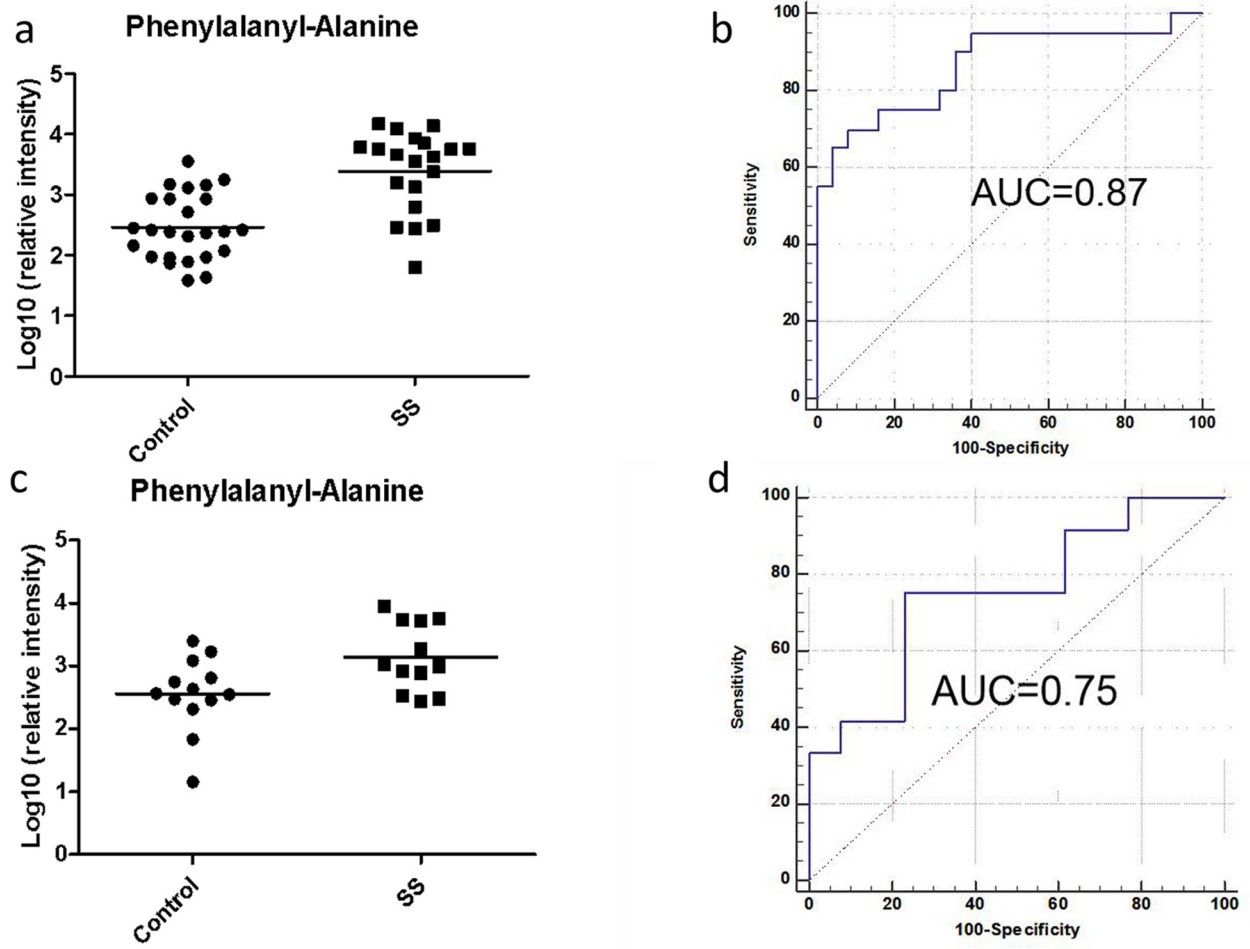

**Fig 5. Potential biomarker using phenylalanyl-alanine for pSS prediction from healthy controls.** (a) Relative intensity of metabolites in pSS patients and controls (test group). (b) ROC curve established using logistic regression (test group). (c) Relative intensity of metabolites in pSS patients and controls (validation group). (d) ROC curve in pSS patients and controls (validation group).

necessary to perform targeted and absolute quantitative analysis to solve this problem, which is our future project.

According to the LC-HRMS spectra of human saliva, we identified a total of 38 metabolites that could be used to significantly differentiate patients with pSS from healthy controls. Our results indicated that the most affected pathways included several amino acid metabolism pathways. Elevated levels of amino acid dipeptides and their products are pathogenic factors for neurological disorders, oxidative stress, and cardiovascular disease. Some amino acids regulate key metabolic pathways that are necessary for maintenance, growth, reproduction, immunity and inflammation [29].

One of the most important results of our study is that the number of metabolites, many of which were dipeptides, derived from tyrosine and phenylalanine metabolic pathways was upregulated in saliva samples. Consistent with our results, phenylalanine concentrations were significantly higher in salivary samples of pSS patients in previous studies, which were analyzed by H-NMR pectroscopy [25]. Phenylalanine hydroxylation to tyrosine, which can synthesize important neurotransmitters and hormones, is carried out by phenylalanine hydroxylase [30]. In recent studies, proteomic analysis revealed that tyrosine-protein

phosphatase nonreceptor type 6 (PTPN6) was dysregulated in saliva and salivary glands in pSS murine models [31]. Receptor tyrosine kinase signaling appears to be disrupted in SS patients, potentially promoting the development of autoantibodies, lymphocytic infiltration, and overt disease [32]. This suggests that tyrosine and phenylalanine play a role in inflammatory injury in this disease.

Moreover, tryptophan metabolism was another significantly upregulated metabolic pathway in our study. Consistent with our results, the products involved in tryptophan metabolism are also changed in the blood and urine samples of pSS patients [33]. An imbalance in the synthesis of tryptophan metabolites has been associated with pathophysiologic mechanisms occurring in neurologic and psychiatric disorders [34, 35], in chronic immune activation [36] and in the immune escape of cancer [37, 38]. Previous results suggested that tryptophan metabolism regulated the immune response in pSS [39].

The results of differential metabolite identification suggested that disorded metabolism of aspartic is also involved in pSS. Aspartic acid is an excitatory amino acid and has strong excitatory effects on neurons. Elevated levels of aspartic acid and phenylalanine in the brain have been linked to behavioral and cognitive problems [40]. Distal sensory and sensorimotor neuropathies are the most common manifestations of peripheral nerve disease in pSS [41]. From our results, we speculate that elevated aspartic acid may be related to nervous system conditions in pSS.

Notably, the level of proline was elevated in saliva from pSS patients in our study, which is consistent with a previous study [24, 25, 33, 42]. Proline metabolism has complex roles in a variety of biological processes, including cell signaling, stress protection, and energy production [43]. Proline metabolic abnormalities affect the citric acid cycle and reduce cell metabolism in an inflammatory state, resulting in cartilage and bone damage [44]. Epstein-Barr virus nuclear antigen (EBNA)-1, a proline-rich sequence, was elevated significantly in patients with systemic lupus erythematosus (SLE) and pSS [45]. Research suggests that inhibiting proline metabolism and transport may be a useful therapeutic strategy against some pathogens.

We noticed that many dipeptides were upregulated. The lysosomal pathway, ubiquitination pathway and caspase pathway are major pathways of protein degradation. Lysosome-associated membrane protein 3(LAMP3) is classically associated with the lysosome, a main organelle central to autophagy. A previous study reported that in SS cases, the expression of LAMP3 was increased, which was related to apoptosis [46]. For the caspase pathway previous studies found that saliva levels of caspase-1 were significantly higher in SS patients, which can induce the production of apoptosis [47]. According to above studies, several important protein degradation pathways in SS were upregulated. Therefore, it was possible that protein degradation was increased in SS, which led to more dipeptides.

In addition to disturbances in amino acid metabolism, our study found abnormalities in purine metabolism. The related metabolites were oxypurinol and 8-hydroxyadenine. Disturbances in purine metabolism can cause gout, which manifests as arthritis due to the crystals produced in hyperuricemia [48]. A previous study found that the mRNA and protein levels of purinergic receptor P2X ligand-gated ion channel 7 (P2X7R) in pSS peripheral blood mononuclear cells were significantly higher than those in normal individuals [49] P2X7R participates in the pathogenesis of pSS. Activation of P2X7 is known to lead to processing and secretion of the proinflammatory cytokines IL-1β and IL-18 from monocytes/macrophages via activation of the NALP3 inflammasome, which is thought to play a role in a spectrum of inflammatory diseases [50]. Upregulation of purine metabolism indicates an inflammatory response in pSS patients.

This caught our attention because carbon fixation was abnormal in our study. This pathway is a cyclic reaction that consumes ATP and NADPH continuously and immobilizes $CO_2$ to

form glucose. However, there are no reports of abnormal carbon cycles in pSS patients. Human cohorts and animal models provide compelling data suggesting the role of the one-carbon cycle in modulating the risk of diabetes, adiposity [51] and fatty liver [52]. Studies have reported that pSS patients exhibit markedly higher prevalence rates of metabolic disorders, such as diabetes and dyslipidemia [53]. Therefore, carbon fixation imbalance may be associated with abnormal lipid metabolism in pSS.

## Conclusion

In conclusion, our study demonstrates that saliva metabolites can be utilized for biomarker discovery in pSS. In our pilot study of pSS saliva metabolic profiling, we were able to distinguish the pSS group from the control group, and panels of metabolites were discovered to have potential value for pSS diagnosis. The potential biomarkers indicated that pSS metabolic disturbance might be associated with inflammatory injury, neurological cognitive impairment, immune response and abnormal lipid metabolism. Overall, our data will benefit the application of the saliva metabolome to disease biomarker discovery, potentially leading to new strategies for assessing the disease.

This work was a pilot study for pSS metabolic biomarker analysis. Further studies should expand the number of pSS patient samples and validate the potential metabolite biomarkers using targedted method, which is our future work. Additionally, further studies should expand the number of pSS patient samples and refine the metabolite profiles of pSS saliva, observe the changes in these biomarkers before and after treatment, and identify biomarkers associated with treatment efficacy, which will hopefully provide an important basis for clinical efficacy evaluation and prognosis prediction in pSS.

## Supporting information

**S1 Fig.** (a) Tight clustering of QC samples indicated good stability of the analysis. (b) Platform 100 times permutation tests for the OPLS-DA model.
(TIF)

**S1 Table. The metabolites related with pSS/control.** Fold change > 1 indicates a relatively higher concentration of the marker present in pSS patients, whereas < 1 indicates a relatively lower concentration of the marker compared with healthy control subjects.
(DOCX)

**S2 Table. Receiver operating characteristic (ROC) analysis of potential pSS biomarkers.** Note: The sensitives and specificities were calculated at their best cutoff points.
(DOCX)

**S1 File. PLOS One clinical studies checklist.**
(PDF)

## Acknowledgments

The authors thank all participants in this study. Study concept and design: Qian Li and Xin You. Patient recruitment: Xin You, Zhen Li, Yue Mu, Chunlan Guo. Data acquisition: Yue Mu, Zhen Li. Data analysis and interpretation: Xiaoyan Liu, Wei Sun. Manuscript writing: Zhen Li, Xiaoyan Liu. Approval of the final version: Wei Sun, Qian Li. All authors reviewed the manuscript before submission.

## Author Contributions

**Conceptualization:** Zhen Li, Qian Li, Wei Sun.

**Data curation:** Yue Mu, Xiaoyan Liu.

**Formal analysis:** Xiaoyan Liu, Wei Sun.

**Funding acquisition:** Qian Li.

**Investigation:** Yue Mu.

**Methodology:** Zhen Li, Chunlan Guo, Qian Li, Wei Sun.

**Project administration:** Yue Mu, Xiaoyan Liu.

**Resources:** Xin You.

**Software:** Xiaoyan Liu, Wei Sun.

**Supervision:** Zhen Li, Qian Li, Wei Sun.

**Validation:** Xiaoyan Liu.

**Visualization:** Xiaoyan Liu.

**Writing – original draft:** Zhen Li.

**Writing – review & editing:** Qian Li, Wei Sun.

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
