## [Decision Letter · Decision Letter 0]

1 Mar 2022

PONE-D-21-33918

Analysis of the saliva metabolic signature in patients with primary Sjögren's syndrome

PLOS ONE

Dear Dr. Li,

Thank you for submitting your manuscript to PLOS ONE. After careful consideration, we feel that it has merit but does not fully meet PLOS ONE’s publication criteria as it currently stands. Therefore, we invite you to submit a revised version of the manuscript that addresses the points raised during the review process.

Please carefully review the comments provided from the reviewer and respond to each statement.  It will be important to consider each statement and make the necessary revisions.  

We look forward to receiving your revised manuscript.

Kind regards,

Timothy J Garrett, PhD

Academic Editor

PLOS ONE

Journal Requirements:

4. Please upload a new copy of Figure 3 as the detail is not clear. Please follow the link for more information: https://blogs.plos.org/plos/2019/06/looking-good-tips-for-creating-your-plos-figures-graphics/" https://blogs.plos.org/plos/2019/06/looking-good-tips-for-creating-your-plos-figures-graphics/

Additional Editor Comments (if provided):

I apologize for the delay in handling this manuscript. It was very difficult to find reviewers for the article. In the end, I have only received 1 review. Please carefully review the comments from the reviewer and provide a response to each point.

Reviewers' comments:

Reviewer's Responses to Questions

**Comments to the Author**

1. Is the manuscript technically sound, and do the data support the conclusions?

Reviewer #1: Partly

2. Has the statistical analysis been performed appropriately and rigorously? 

Reviewer #1: No

3. Have the authors made all data underlying the findings in their manuscript fully available?

Reviewer #1: Yes

4. Is the manuscript presented in an intelligible fashion and written in standard English?

Reviewer #1: No

5. Review Comments to the Author

Reviewer #1: PONE-D-21-33918

Li et al. studied that saliva metabolic profiling to explore changes in metabolite profiles and identify unique biomarkers for Sjögren’s syndrome (pSS). Although, this is an interesting work but there several issues that need to be addressed.

Major points:

1. In abstract section, objective need to be specific for biomarker analysis for Sjögren’s syndrome (pSS)

2. Sample were centrifuged at 1500g for 15min, what is the basis for doing this. Saliva contains lots of other matrix including food particles. How does it helps to get rid of these contaminates?

3. Is there any pre-normalization process for saliva? If not please explain how it conducted and how it will control the sample variation?

4. In several places, citation was not provided to locate the proper information. For example – “Raw data files were processed by Progenesis QI (Version 2.0, Nonlinear Dynamics) software based on a previously published identification strategy.” – there should be a reference that need to be add here.

5. MetaboAnalyst 5.0 is currently up-to-date version, why did author choose MetaboAnalyst 3.0 instead of latest version?

6. It was mentioned that QC was done by specialist, this is very generalized word. It need to be clearly stated that what is this specialist mean here?

7. In Figure 2A, author talked about the PCA analysis but in Figure 2B oPLS-DA, it is important to mention the PLS-DA Q2 model value to show the robustness of clustering for differential metabolite analysis.

8. 676 features with a P value < 0.05 were submitted for pathway enrichment analysis using the Mummichog algorithm, are these upregulated or downregulated or combined features? No information mentioned here.

9. Fig 3 claimed all the pathways are upregulated. It can be only possible if upregulated metabolites used for pathway analysis. It is confusing to read and connect with the biology otherwise. Please revisit the data and state this clearly.

10. Figure 4 heatmap clearly represent the challenge of saliva normalization, although it is difficult to do proper pre-normalization of saliva, but it will be greatly appreciated if the author could explain this limitation carefully.

11. Figure 4 is currently provided as customized and supervised manner. It is important to provide this figure with sample vs feature by unsupervised cluster so that top feature can be better understood.

12. Author claimed so many dipeptides are upregulated. Does it mean elevated protein degradation with the disease? Further follow up is missing in the discussion.

13. Targeted validation of claimed biomarker is missing.

Minor points:

1. Proper punctuation missing for several sentences: for example

“Saliva is a complex fluid containing a variety of metabolites, proteins, mRNAs, DNAs, enzymes, hormones, antibodies, antimicrobial constituents, growth factors and other molecules that may be associated with disease phenotypes [10, 11],” – it shouldn’t be comma. Likewise, several places there are missing of proper punctuation.

2. Overall, most of the figures need to provide with high resolution. Their current format is obscure.

6. PLOS authors have the option to publish the peer review history of their article (what does this mean?). If published, this will include your full peer review and any attached files.

Reviewer #1: No

---

## [Author Response · Author response to Decision Letter 0]

9 Apr 2022

Dear Editor, 

Thank you for carefully reviewing our manuscript previously titled “Analysis of the saliva metabolic signature in patients with primary Sjögren's syndrome” for possible publication in PLoS One. We are grateful to you and your reviewers for their constructive critique. We have revised the manuscript, highlighting our revisions in yellow, and have attached point-by-point responses detailing how we have revised the manuscript in response to the reviewers' comments below.

Thank you for your consideration and further review of our manuscript. Please do not hesitate to contact us with any further questions or recommendations.

Yours Sincerely,

Qian Li

Reviewer Comments:

Reviewer #1:

1. In abstract section, objective need to be specific for biomarker analysis for Sjögren’s syndrome (pSS)

Response: Thanks very much for your comments. As you suggested,abstract section had been added to the revised manuscript as follows. 

Abstract section: “

Background

The saliva metabolome has been applied to explore the disease biomarkers. In this study we aim to characterize the metabolic profile of Primary Sjögren’s Syndrome (pSS) patients and explore the metabolomic biomarkers.” (Page 2, Line 2-5)

2. Sample were centrifuged at 1500g for 15min, what is the basis for doing this. Saliva contains lots of other matrix including food particles. How does it helps to get rid of these contaminates?

Response: Thanks very much for your comments. In our study, the unstimulated saliva sample collection was conducted at the same time of day (09:00–11:00 a.m.) to overcome any circadian influences. The subjects were requested to refrain from eating, drinking, smoking or oral hygiene procedures for at least 1.5 hr prior to saliva collection, and to rinse their mouth with water just prior to sample collection. This operation can minimize matrix including food particles in saliva. Then the samples were centrifuged at 1500 × g for 15 min at 4°C to remove exshed cell debris and residue, which might be helpful to get rid of food contaminates. 

Previous investigators also used the similar saliva collection process (ref: Metabolomics analysis of saliva from patients with primary Sjögren's syndrome. Clin Exp Immunol. 2015;182(2):149-53.). We have added the above statement and reference to the manuscript as follows:

Materials and methods section: “ The samples were transported to the laboratory on ice and centrifuged at 1500 × g for 15 min at 4°C to remove shed cell debris and residue, and then the supernatants were frozen immediately and stored at -80°C until analysis[23].”(Page 7, Line 8-10)

3. Is there any pre-normalization process for saliva? If not please explain how it conducted and how it will control the sample variation? 

Response: Thanks very much for your comments. Reliable identification of features distinguishing biological groups in salivary metabolite fingerprints requires the control of total metabolite abundance. Normalizations to one index values (for example, creatinine for urine metabolomics), osmolality, and total compound (useful MS signals) were the commonly used normalization techniques for overcoming sample variability in fluids metabolomics. Previous researches have compared the effect of different normalization method on urine metabolomics study. It showed that the best performance was normalization to the total compound. The approach does not only correct for different urinary output, but it also accounts for injection variability (Anal Bioanal Chem (2016) 408:8483-8493. DOI 10.1007/s00216-016-9974-1). S, Journal of Chromatography B. 2009, 877(5–6),). 

According to above references, in present study, we normalized the data using the method of “Normalize to the total compound”. The detailed normalization method was as follows:

1) Normalization reference (the 'target'): one run is automatically selected as the normalization reference.

2) Log10 ratio calculation: for every run, a ratio can be taken for the value of the compound ion abundance in that run to the value in the normalization reference. This ratio calculation removes the influence of absolute abundance from the process, which is a major advantage over total-abundance-based methods.

3) Scalar estimation in log space: the next step is to center the log10 ratio distributions onto that of the normalization reference in each case. 

4) Scalar application: once the scalar has been derived in log space and then returned to an ‘abundance-space ratio’, it can be applied to all values in the sample run being normalized, and this completes the process.”

We have added above statements in the manuscript (Page 8, Line 18; Page18, Line 13-21) 

4. In several places, citation was not provided to locate the proper information. For example – “Raw data files were processed by Progenesis QI (Version 2.0, Nonlinear Dynamics) software based on a previously published identification strategy.” – there should be a reference that need to be add here.

Response: Thanks very much for your comments. We have added the references in the text “

Statistical Analysis

Raw data files were processed by Progenesis QI (Version 2.0, Nonlinear Dynamics) software based on a previously published identification strategy, which included sample alignment, peak picking, peak grouping, deconvolution, normalization by total compounds, and final information export (ref:An intelligentized strategy for endogenous small molecules characterization and quality evaluation of earthworm from two geographic origins by ultra-high performance HILIC/QTOF MS(E) and Progenesis QI.Anal Bioanal Chem. 2016 May;408(14):3881-90.) ” (Page 8 ,line 15-19)

5. MetaboAnalyst 5.0 is currently up-to-date version, why did author choose MetaboAnalyst 3.0 instead of latest version?

Answer: Thanks for the reviewer’s suggestion. The data was analyzed MetaboAnalyst 3.0. MetaboAnalyst version 5.0 aimed to narrow the gap from raw data to functional insights for global metabolomics based on high-resolution mass spectrometry (HRMS). And we re-analyzed the function pathway using MetaboAnalyst 5.0 and got the similar results to those using MetaboAnalyst 3.0 (FigR1). 

In the revised manuscript we have updated the results (Figure 3).

Fig R1.Analysis results using MetaboAnalyst 3.0 (A) and MetaboAnalyst 5.0 (B). 

6. It was mentioned that QC was done by specialist, this is very generalized word. It need to be clearly stated that what is this specialist mean here? 

Response: Thanks very much for the reviewer’s suggestion. The “specialist”means the professionals these who have received professional LC/MS/MS analysis training. We have revised “specialist” as “professionals ” (Page 11 , line 17)

7. In Figure 2A, author talked about the PCA analysis but in Figure 2B oPLS-DA, it is important to mention the PLS-DA Q2 model value to show the robustness of clustering for differential metabolite analysis.

Response: Thanks for the reviewer’s suggestion. We have added the Q2 value and the CV-ANOVA p value of OPLS-DA model to validate the robustness of the model. And further permutation test was performed to ensure no overfitting of this model (intercept of Q2 = - 0.261). 

We have added these statements in the mansucript “the OPLS-DA model achieved better separation for differential metabolite selection (R2X=0.413, R2Y=0.932, Q2=0.46, CV-ANOVA, p = 0.00070, Fig 2B). Permutation tests (100 times) were performed to confirm the stability and robustness of the supervised models presented in this study (intercept of Q2 = - 0.261, Supplementary Figure 1B).” (Page 12 , line 6-10)

8. 676 features with a P value < 0.05 were submitted for pathway enrichment analysis using the Mummichog algorithm, are these upregulated or downregulated or combined features? No information mentioned here.

Response: Thanks for the reviewer’s suggestion. We have added the information in the manuscript “Overall, 676 features with a P value < 0.05 (351 upregulated and 325 downregulated features) were submitted for pathway enrichment analysis using the Mummichog algorithm.” (Page 12 , line 11)

9. Fig 3 claimed all the pathways are upregulated. It can be only possible if upregulated metabolites used for pathway analysis. It is confusing to read and connect with the biology otherwise. Please revisit the data and state this clearly.

Response: Thanks for the reviewer’s suggestion. We have carefully checked the change trend of differential features enriched in these pathways. Metabolites involved in Tryptophan metabolism, Aspartate and asparagine metabolism showed up-regulated. And metabolites involved in Carbon fixation showed down-regulated. 

We have revised the statements in the manuscript “Metabolites involved in tryptophan metabolism and aspartate and asparagine metabolism were upregulated. Metabolites involved in carbon fixation were downregulated(Fig 3)” (Page 12 ,line 15-17)

10. Figure 4 heatmap clearly represent the challenge of saliva normalization, although it is difficult to do proper pre-normalization of saliva, but it will be greatly appreciated if the author could explain this limitation carefully.

Response: Thanks for the reviewer’s suggestion. Pre-normalization of saliva is indeed a difficult problem. 

The concentrations of endogenous metabolites in salivary vary widely and normalizing for these effects is necessary. Reliable identification of features distinguishing biological groups in salivary metabolite fingerprints requires the control of total metabolite abundance. Normalizations to one index values (for example, creatinine for urine metabolomics normalization), osmolality, and total compound (useful MS signals) were the commonly used normalization techniques for overcoming sample variability in fluids metabolomics (Warrack et al., 2009). Researches have compared the effect of different normalization method on urine metabolomics study. It showed that the best performance was from normalization to the total compound. The approach does not only correct for different urinary output, but it also accounts for injection variability (Warrack et al., 2009; Vogl et al., 2016). Thus, in present study, we normalized the data using the method of “Normalize to the total compound”. However, it cannot completely solve the problem of pre-normalization. It is necessary to perform targeted and absolute quantitative analysis to solve this problem, which is our future project. 

We have added above statements in the discussion section

“Whole saliva is a complex fluid containing a variety of substances, and changes in the levels of these substances are related to the pathogenesis of various diseases. However, the function of the salivary glands in patients with pSS is severely weakened, resulting in the volume of saliva being too small. Patients' whole saliva up to 2 ml was collected for analysis over time to obtain a pre-normalization of saliva. 

Reliable identification of features distinguishing biological groups in salivary metabolite fingerprints requires the control of total metabolite abundance. Normalizations to one index values (for example, creatinine for urine metabolomics), osmolality, and total compound (useful MS signals) were the commonly used normalization techniques for overcoming sample variability in fluids metabolomics. Previous researches have compared the effect of different normalization method on urine metabolomics study. It showed that the best performance was normalization to the total compound. The approach does not only correct for different urinary output, but it also accounts for injection variability (REF:Evaluation of dilution and normalization strategies to correct for urinary output in HPLC-HRTOFMS metabolomics.). In present study, we normalized the data using the method of “Normalize to useful MS signals” to reduce sample variation. However, it cannot completely solve the problem of pre-normalization. It is necessary to perform targeted and absolute quantitative analysis to solve this problem, which is our future project. ” (Page 18, line 8-21)

11. Figure 4 is currently provided as customized and supervised manner. It is important to provide this figure with sample vs feature by unsupervised cluster so that top feature can be better understood.

Response: Thanks for the reviewer’s suggestion. we have plotted this figure with sample vs feature by unsupervised cluster as following, and the results showed the metabolites could distinguish the pSS subjects and healthy controls. 

Fig R4 Relative intensity of differential metabolites in pSS subjects and healthy controls

12. Author claimed so many dipeptides are upregulated. Does it mean elevated protein degradation with the disease? Further follow up is missing in the discussion.

Response: Thanks very much for your comments. We have added a note to the discussion，

“We noticed that many dipeptides are upregulated. Lysosomal pathway, ubiquitination pathway, and caspase pathway are major pathways of protein degradation. Lysosome-associated membrane protein 3(LAMP3) is classically associated with the lysosome, a main organelle central to autophagy. Previous study reported that In Sjogren's syndrome cases, the expression of LAMP3 was increased, which was related to apoptosis.（ref: LAMP3 induces apoptosis and autoantigen release in Sjögren's syndrome patients. 2020 Sep 16;10(1):15169. ). For caspase pathway previous studies found that saliva levels of caspase-1 were significantly higher in SS patients , which can induce the production of apoptosis（ref: Saliva levels of caspase-1, TNF-α, and IFN-γ in primary Sjögren's syndrome: oral mucosal abnormalities revisited. Turk J Med Sci, 2018 Jun 14;48(3):554-559.). According to above studies, several important protein degradation pathways in SS were up-regulated. Therefore, it was possible that protein degradation was increased in SS, which lead to more dipeptides.”(Page 21 ,line 6-15)

13. Targeted validation of claimed biomarker is missing.

Response: Thanks for the reviewer’s suggestion. This work was a pilot study for SS metabolic biomarker analysis. We totally agree with the reviewer’s suggestion that targeted validation of claimed biomarker with more samples is necessary. We are trying to collect samples from multiple centers, and to validate the metabolic biomarkers. We will present validation work in the future. We have added above statements in the manuscript “This work was a pilot study for SS metabolic biomarker analysis. Further studies should expand the number of pSS patient samples and validate the potential metabolite biomarkers using targedted method, which is our future work.”(Page23, Line 2-4) 

Minor points:

1. Proper punctuation missing for several sentences

Response: Thanks very much for your comments. We have carefully examined the punctuation and the letter case.

2. Overall, most of the figures need to provide with high resolution. Their current format is obscure.

Response: Thanks very much for your comment. We have re-uploaded the figures with higher definition and resolution and hope these figures will meet the requirement.

---

## [Editor Report · Decision Letter 1]

18 May 2022

Analysis of the saliva metabolic signature in patients with primary Sjögren's syndrome

PONE-D-21-33918R1

Dear Dr. Li,

We’re pleased to inform you that your manuscript has been judged scientifically suitable for publication and will be formally accepted for publication once it meets all outstanding technical requirements.

Kind regards,

Timothy J Garrett, PhD

Academic Editor

PLOS ONE
---

## [Editor Report · Acceptance letter]

23 May 2022

PONE-D-21-33918R1 

Analysis of the saliva metabolic signature in patients with primary Sjögren's syndrome 

Dear Dr. Li:

I'm pleased to inform you that your manuscript has been deemed suitable for publication in PLOS ONE. Congratulations! Your manuscript is now with our production department. 

Kind regards, 

on behalf of

Dr. Timothy J Garrett 

Academic Editor

PLOS ONE